# Blastocyst Transplantation Strategies in Women of Different Ages

**DOI:** 10.3390/jcm12041618

**Published:** 2023-02-17

**Authors:** Dandan Yang, Menghan Chai, Ni Yang, Han Yang, Xingxing Wen, Jing Wang, Yongqi Fan, Yunxia Cao, Zhiguo Zhang, Beili Chen

**Affiliations:** 1Reproductive Medicine Center, Department of Obstetrics and Gynecology, The First Affiliated Hospital of Anhui Medical University, No. 81 Meishan Road, Hefei 230032, China; 2NHC Key Laboratory of Study on Abnormal Gametes and Reproductive Tract, Anhui Medical University, No. 81 Meishan Road, Hefei 230032, China; 3Department of Biomedical Engineering, Anhui Medical University, No. 81 Meishan Road, Hefei 230032, China

**Keywords:** single blastocyst transfer, double blastocyst transfer, live birth rate, multiple birth rate, preterm birth rate

## Abstract

(1) Background: Single blastocyst transfers (SBT) and double blastocyst transfers (DBT) are widely used in clinical practice. The objective of this study was to investigate the application of these two strategies in women of different ages. (2) Methods: Analysis was carried out on 5477 frozen embryo transfer cycles of women in different ages. All the cycles were divided into three groups according to the age of the included women: <35, 35–39, and >39. The live birth rate (LBR) and multiple birth rate (MBR) between the SBT and DBT among these groups, respectively, were compared. (3) Results: In the women < 35 group, the LBR was similar in SBT and DBT, but the MBR was higher in DBT than SBT. In women 35–39, with >10 oocytes retrieved, the MBR in DBT was significantly higher than SBT, but there was no significant difference in LBR between the two groups; with ≤10 oocytes retrieved, the LBR in DBT were significantly higher than SBT, but the MBT was not significantly different between the two groups. In women > 39, the LBR and MBR were lower in the SBT than DBT, but the differences were not statistically significant. (4) Conclusions: Selective SET is appropriate for most young women, whereas older women are advised to make individualized choices based on the number of oocytes retrieved and blastocyst quality.

## 1. Introduction

The wide application of assisted reproductive technology (ART) has made great progress in the treatment of infertile couples. A single delivery of a healthy child is the ideal outcome of ART. However, multiple embryos were transferred to increase the clinical pregnancy rate and live birth rate per embryo transfer cycle over the past several decades, which has led to higher rates of multiple pregnancy and a higher incidence of pregnancy complications. In the past 30 years, the overall rate of twin pregnancy increased by 50~70%, and the rate of triplet pregnancy increased by 400% [1]. Multiple pregnancies pose a serious threat to maternal and infant health, and numerous studies have shown a significant correlation between the number of pregnancies and adverse pregnancy outcomes [2]. Twin and multiple pregnancies are associated with an increased incidence of maternal and neonatal complications, including preterm birth, intrauterine growth restriction, low birth weight infants, cerebral palsy, cognitive impairment, and developmental delays [3,4,5,6]. In addition, the immaturity of various organs in preterm infants can make the care of preterm infants a difficult problem for obstetricians [7]. Meanwhile, the cost of care and treatment increases the financial burden on patients. Therefore, it is particularly important to reduce the occurrence of multiple pregnancy while ensuring a high clinical pregnancy rate. Although multiple pregnancy reduction can reduce the number of fetuses, the procedure may result in the loss of all fetuses and still has many adverse effects [8]. In addition, not all patients are willing to undergo such surgery. In recent years, selective single embryo transfer (eSET) was considered to be the most effective prophylactic measure to address multiple pregnancies of medical origin and to improve maternal and infant safety [9,10]. In 2018, it was even suggested that eSET is suitable for all populations [11]. However, the choice of embryo transfer strategy remains controversial among different age groups.

At present, single blastocyst transfer (SBT) and double blastocyst transfer (DBT) are widely used in clinical practice. The objective of this study was to investigate the application of these two strategies in women of different age groups who first underwent a frozen blastocyst transplantation cycle in the Reproductive Center of the First Affiliated Hospital of Anhui Medical University from October 2018 to October 2021.

## 2. Materials and Methods

### 2.1. Ethics Statement

This study was initiated and conducted after receiving approval from the Committee of Medical Ethics from Anhui Medical University (No. 2017002).

### 2.2. Study Subjects

This was a retrospective, single-center cohort study that included all women without a fresh embryo transplant in our center who underwent their first resuscitation cycle blastocyst transfer from October 2018 to October 2021. A total of 5477 cycles were enrolled in the study after excluding patients with various uterine malformations, adenomyosis, severe endometriosis, recurrent miscarriages, and preimplantation genetic diagnoses of the blastocyst.

### 2.3. Superovulation, Embryo Culture, and Transfer

In this study, the patients received either a long or short regimen used routinely in our center for ovarian induction, and the detailed process is described in our previous article [12]. When 1–2 dominant follicles with a diameter of 18 mm, or more than 2 dominant follicles with a diameter of 17 mm were presented, 5000–10,000 IU of HCG (Lizhu Pharmaceutical Trading Co., Ltd., Zhuhai, China) were injected. In addition, oocyte retrieval was completed after 36 h with a transvaginal ultrasound.

Under a microscope, the oocytes in follicular fluid were picked up and cultured for a short time in vitro in fertilization medium (COOK, Bloomington, IN, USA), and they then underwent ICSI or IVF insemination based on sperm quality. At 14–18 h after IVF/ICSI insemination, the fertilized oocytes were assessed by the presence of pronucleus, and then cleavage embryo culturing was continued for two days in cleavage medium (COOK, USA) with a subsequent two or three days of blastocyst culturing in blastocyst medium (COOK, USA). Finally, the obtained blastocysts were selected and cryopreserved in −196 °C liquid nitrogen using the vitrification method (KITAZATO, Fuji, Japan) to be followed by thawed embryo transfer.

Three months later, one or two warmed blastocysts were transferred into the uterus under ultrasound guidance. More operation procedures are described in the literature [13].

### 2.4. The Selection Criteria for “SET or DET”

According to Chinese experts’ consensus, elective SBT is recommended for all couples in the first transplantation cycle, whereas SBT is strongly recommended for patients with a history of cesarean section. In addition, personal wishes and demands should be taken into account in the determination of the number of blastocysts transplanted. After full communication with patients and careful assessment of patients’ basic conditions, DBT was performed in patients who strongly requested and were willing to accept all risks caused by multiple pregnancy.

### 2.5. Indicator Assessment

In this study, high-quality embryos at the cleavage stage were defined as embryos with 7–9 blastomeres at equal size on day 3 with no or less than 15% Fragmentation. “High-quality blastocysts” refers to embryos whose grades were above 3BB on day 5 or above 4BB on day 6 as determined using the blastocyst grading system according to Gardner’s criteria [14].

Two weeks after embryo transfer, a positive serum β-hCG level (≥10 IU/L) was defined as a biochemical pregnancy. Thirty days after embryo transfer, the presence of a gestational sac on an ultrasound scan was defined as clinical pregnancy. In addition, preterm birth was defined as delivery between 28 weeks and less than 37 weeks of gestation. Preterm birth was further subdivided into very early preterm (<28 weeks), early preterm (28–33 weeks), and late preterm (34–36 weeks) [15].

Biochemical pregnancy rate = Number of biochemical pregnancies cycles/Number of transplant cycles × 100%; Clinical pregnancy rate = Number of clinical pregnancy cycles/Number of transfer cycles × 100%; Multiple gestational sacs rate = Number of multiple gestational sacs cycles/Number of intrauterine pregnancy cycles × 100%; Live birth rate = Number of live birth cycles/Number of transplant cycles × 100%; Multiple birth rate = Number of multiple birth cycles/Number of live birth cycles × 100%; Preterm birth rate = Number of preterm birth cycles/Number of live birth cycles × 100%.

### 2.6. Statistical Analysis

Statistical analysis was carried out by using GraphPad Prism 8.0 and SPSS 23.0 software. The measurement data were expressed in terms of mean ± standard deviation, and the enumeration data were expressed as a percentage. The measurement data were tested by using two independent sample *t*-tests, and the χ^2^ test was used to enumerate data. *p* < 0.05 was considered to be statistically significant. Binary logistic regression analysis was used to verify the independent influencing factors of the live birth rate and preterm birth rate.

## 3. Results

### 3.1. Clinical Outcomes of SBT and DBT

Among the 5477 cycles included in this analysis, there were 3323 cycles of SBT and 2154 cycles of DBT. The comparison between the two groups showed that the rates of biochemical pregnancy, clinical pregnancy, and live birth in the SBT and DBT groups were similar. However, the rate of multiple gestational sacs, multiple birth, and preterm birth in the DBT group were significantly higher than those of the SBT group (Table 1, *p* < 0.05).

Furthermore, a breakdown of the preterm birth data revealed that the mean gestational age and the rates of late, early, and very early preterm births were similar in both groups (*p* > 0.05), with the majority of preterm births being late preterm and only a few cases of very early preterm births (Table 1, *p* < 0.05).

### 3.2. Clinical Outcomes of SBT versus DBT in Different Age Groups

Because the age of DBT was significantly higher than that of the SBT group, and because age was an important factor affecting the outcome of transplantation, all transfer cycles were divided into three age groups according to the age of patients: Young (under 35 years old), Medium (35 to 39 years old), and Aged (over 39 years old). Table 2 shows the clinical outcomes of the different blastocyst transfer protocols among three age groups: young SBT (Group A), young DBT (Group B), medium SBT (Group C), medium DBT (Group D), aged SBT (Group E), and aged DBT (Group F).

In women aged < 35 years, the rates of biochemical pregnancy, clinical pregnancy, and live birth were similar in the SBT and DBT groups, but the rates of multiple gestational sacs, multiple births, and preterm births were significantly higher in the DBT group than in the SBT group (*p* < 0.05).

In women aged 35–39 years, the rates of biochemical pregnancy, clinical pregnancy, and live birth in the DBT group were significantly higher than those of the SBT group, and the rates of multiple gestational sacs, multiple birth, and premature birth were also significantly higher than those of SBT group (*p* < 0.05).

In women aged over 39 years, there were no significant differences in the rates of biochemical pregnancy, clinical pregnancy, live birth, and preterm birth between SBT and DBT in transplant cycles (*p* > 0.05). The rate of multiple births was lower in the SBT group (Group E) than in the DBT group (Group F), but the differences were not statistically significant (*p* > 0.05).

### 3.3. Logistic Regression of Live Birth and Preterm Birth among Women Aged 35–39 Years

For women aged 35–39 years, the advantages and disadvantages of both SBT and DBT are clear, and as it is difficult to generalize, a more detailed analysis of patients in this age group is required. To investigate which factors influence live birth among women aged 35–39 years, we divided 792 women into a successful birth group (MS) and an unsuccessful birth group (MU), and the basic characteristics of the two groups are reported in Table 3. We found significant differences between the MS and MU groups in the measures of primary infertility, the number of oocytes retrieved, the rate of fertilization, the percentage of patients with at least one high-quality blastocyst, and the proportion of SBT. To further explore the factors influencing preterm birth in this age group, 308 women who delivered successfully were divided into the full-term (MF) and preterm (MP) groups. Univariate analysis showed that the two groups only had a significant difference in the number of transplanted blastocysts (Table 3).

Subsequently, a logistic regression analysis was carried out with live birth and preterm birth as dependent variables and indicators with statistical differences as independent variables. The results are shown in Table 4. The number of oocytes retrieved, the presence or absence of high-quality blastocysts, and the number of blastocysts transferred significantly correlated with live birth among women aged 35 to 39 (*p* < 0.05). There was a 1-fold increase in the live birth rate for each additional oocyte retrieved, a 5-fold increase in live birth rate with high-quality blastocysts compared to no high-quality blastocysts, and a 1.5-fold increase in live birth rate with DBT compared to SBT. In addition, the number of blastocysts transferred was significantly associated with preterm birth (*p* < 0.05), and DBT showed a 2.7-fold increase in the preterm birth rate compared to SBT.

### 3.4. Clinical Outcomes of SBT and DBT in Women Aged 35–39 Years with Different Numbers of Oocytes Retrieved

To further clarify the strategy of blastocyst transfer in this age group, women aged 35–39 years with at least one high-quality blastocyst were grouped according to the number of oocytes retrieved (L: oocytes ≤ 10; H: oocytes > 10) and the number of blastocysts transferred. The clinical outcomes of the four treatment cycles in the LSBT (Group G), LDBT (Group H), HSBT (Group I), and HDBT (Group J) groups are shown in Table 5. In women with ≤10 oocytes retrieved, the rates of biochemical pregnancy, clinical pregnancy, multiple gestational sacs, and live birth in the DBT group were significantly higher than those in the SBT group (*p* < 0.05), but the rates of multiple births and preterm birth were not significantly different between the two groups (*p* > 0.05). Among the women with more than 10 oocytes, the rates of biochemical pregnancy, clinical pregnancy, multiple gestational sacs, multiple birth, and preterm birth in the DBT group were significantly higher than those in the SBT group (*p* < 0.05), but there was no significant difference in the live birth rate between the two groups (*p* > 0.05).

## 4. Discussion

This study had a large sample size of 5477 patients undergoing blastocyst transfer cycles to assess whether clinical outcomes differed between SBT and DBT at different age segments. Due to the selection criteria for the number of blastocysts transplanted in our center, the proportion of patients with a history of cesarean section in DBT group was much lower than that of the SBT group (2.92% vs 15.53%). However, the rate of preterm birth in DBT group was significantly higher than that of SBT group (34.77% vs 13.15%). It is believed that having a history of cesarean section had little effect on the occurrence of preterm birth and no selective bias on the subsequent analysis.

In our current analysis, we indicated that although the clinical outcomes of single or double blastocysts were similar in young women receiving embryo transfer for the first time (age < 35), the DBT group had significantly higher rates of multiple pregnancy and preterm birth. Therefore, for young women undergoing the blastocyst transplant for the first time, selective SBT was undoubtedly the most appropriate choice. This is consistent with the 2017 guidelines of the American Society for Reproductive Medicine (ASRM) that recommend selective SBT for all embryos, whether fresh or thawed, in the first transplant cycle of a patient [16].

For older women, the situation is more complicated. The age-related decline in the ovarian reserve not only reduces the number of oocytes retrieved but also reduces embryo implantation and pregnancy rates [17], and there are no treatments that significantly improve embryo quality in advanced age. Therefore, a reliable embryo selection strategy is key to improving pregnancy rates and safety. According to the latest guidelines published by the ASRM [18], two blastocysts can be selected for transfer in patients of advanced age, but one high-quality blastocyst is preferred. Although many studies have shown no statistical difference in the planting rate between SBT and DBT, DBT may be better at improving the live birth rate [19,20,21]. It was suggested that the selective SET policy applies to women in the 36–39 age group, but high pregnancy rates were observed in women with high-quality embryos [22], which is consistent with our findings. In women 35–39, in addition to the number of blastocysts transferred, the number of oocytes retrieved and the presence of high-quality blastocysts also influenced the live birth rate. Thus, the clinical decision between SET and DET should be based not only on the age of the woman but also on other prognostic features of the treatment cycle, such as the number of oocytes retrieved and the presence of high-quality embryos. It was shown that the number of oocytes retrieved is independently associated with the live birth rate after IVF/ICSI [23,24]. In this study, the live birth rate of DBT was significantly higher than that of SBT in women of this age group with ≤10 oocytes retrieved, and there were no significant differences in the rates of multiple birth and preterm birth between the two groups. In contrast, there was no significant difference in the live birth rate between SBT and DBT in women with >10 oocytes retrieved. Therefore, we believe that the SBT strategy is feasible, safe, and effective in women aged 35–39 years for patients with >10 oocytes retrieved. Conversely, for patients with ≤10 oocytes retrieved or no high-quality blastocysts, DBT is recommended to improve the live birth rate.

In women aged over 39 years, DBT did not increase the rate of multiple pregnancy or preterm birth while increasing the rate of live birth, which is consistent with previous results in the literature [25]. Therefore, DBT is supported in women at this age.

## 5. Conclusions

In conclusion, this study suggests that selective SET is appropriate for young women (under 35 years), which increases the safety of ART treatment without decreasing the live birth rate and minimizes the health risks faced by these women. However, due to the age-related decline in embryonic development potential, the treatment effect of ART will also decline in women with advanced age. Therefore, we do not recommend selective SBT for every woman over 35 years of age; rather, we recommend that these women make an individual choice based on the number of oocytes retrieved and blastocyst quality. In addition, this study is a retrospective study, which means it inevitably has a certain selection bias. Although the inclusion and exclusion criteria were strictly stipulated, there are still certain limitations in the screening of enrolled patients. Future multi-center and multi-book prospective studies are needed to further investigate the effect of the number of blastocyst transplants on clinical pregnancy outcomes.

## Figures and Tables

**Table 1 jcm-12-01618-t001:** Clinical outcome of single blastocyst transfer group (SBT) versus double blastocyst transfer group (DBT).

Index	SBT	DBT	*p*-Value
Total cycles (n)	3323	2154	-
Female age (years)	30.06 ± 4.54	31.86 ± 4.53	<0.0001 *
Biochemical pregnancy rate (%)	64.91 (2157/3323)	65.37 (1408/2154)	0.7297
Clinical pregnancy rate (%)	59.92 (1991/3323)	60.86 (1311/2154)	0.4838
Multiple gestational sacs rate (%)	1.01 (20/1977)	45.01 (586/1302)	<0.0001 *
Live birth rate (%)	46.22 (1536/3323)	48.47 (1044/2154)	0.104
Multiple birth rate (%)	1.95 (30/1536)	34.87 (364/1044)	<0.0001 *
Preterm birth rate (%)	13.15 (202/1536)	34.77 (363/1044)	<0.0001 *
Gestational age of preterm infants (weeks)	34.41 ± 1.96	34.48 ± 1.97	0.6914
Late preterm rate (%)	75.74 (153/202)	78.51(285/363)	0.4497
Early preterm rate (%)	23.76 (48/202)	20.66 (75/363)	0.3919
Very early preterm rate (%)	0.5 (1/202)	0.83 (4/363)	>0.9999

SBT: single blastocyst transfer; DBT: double blastocyst transfer. *: *p* < 0.05, difference was statistically significant.

**Table 2 jcm-12-01618-t002:** Cycle outcomes of the six study groups.

Index	(A) YSBT	(B) YDBT	(C) MSBT	(D) MDBT	(E) ASBT	(F) ADBT	*p*-Value (A vs. B)	*p*-Value (C vs. D)	*p*-Value (E vs. F)
Total cycles (n)	2798	1616	394	398	131	140	-	-	-
Biochemical pregnancy rate (%)	68.55 (1918/2798)	69.06 (1116/1616)	51.02 (201/394)	61.31 (244/398)	29.01 (38/131)	34.29 (48/140)	0.7245	0.0035 *	0.3509
Clinical pregnancy rate (%)	63.37 (1773/2798)	65.10 (1052/1616)	46.70 (184/394)	55.28 (220/398)	25.95 (34/131)	27.86 (39/140)	0.2481	0.0158 *	0.7242
Multiple gestational sacs rate (%)	1.02 (18/1764)	47.52 (498/1048)	1.10 (2/181)	37.5 (81/2216)	0.00 (0/32)	18.42 (7/38)	<0.0001 *	<0.0001 *	0.0133 *
Live birth rate (%)	49.57 (1387/2798)	52.29 (845/1616)	34.01 (134/394)	43.72 (174/398)	11.45 (15/131)	18.57 (26/140)	0.0818	0.0051 *	0.1021
Multiple birth rate (%)	1.87 (26/1387)	37.99 (321/845)	2.99 (4/134)	23.56 (41/174)	0.00 (0/15)	7.69 (2/26)	<0.0001 *	<0.0001 *	0.5244
Preterm birth rate (%)	12.91 (179/1387)	35.98 (304/845)	14.93 (20/134)	32.18 (56/174)	20.00 (3/15)	11.54 (3/26)	<0.0001 *	0.0005 *	0.7797

Y: Young (aged < 35); MSBT: Medium (aged 35–39); ASBT: Aged (aged ≥ 40); SBT: single blastocyst transfer; DBT: double blastocyst transfer, *: *p* < 0.05, difference was statistically significant.

**Table 3 jcm-12-01618-t003:** Univariate analysis in the medium group (women aged between 35 and 39) with live birth or preterm birth as the dependent variable.

Index	MS	MU	MF	MP	*p*-Value (MS vs.MU)	*p*-Value (MF vs. MP)
Total cycles (n)	308	484	232	76	-	-
Female age (years)	36.56 ± 1.42	36.75 ± 1.42	36.56 ± 1.38	36.54 ± 1.54	0.0650	0.9115
Primary infertility rate (%)	28.90 (89/308)	21.69 (105/484)	29.31 (68/232)	27.63 (21/76)	0.0216 *	0.7793
BMI (kg/m^2^)	23.06 ± 3.27	23.16 ± 3.18	22.93 ± 3.29	23.44 ± 3.20	0.6626	0.2417
bFSH (IU/L)	7.94 ± 2.90	7.90 ± 2.73	8.02 ± 2.98	7.67 ± 2.64	0.8434	0.3621
bLH (IU/L)	4.94 ± 2.43	4.93 ± 2.82	5.02 ± 2.48	4.71 ± 2.26	0.9358	0.3335
bE2 (pmol/L)	183.3 ± 142.9	194.8 ± 168.4	185.6 ± 154.5	176.1 ± 100.2	0.3188	0.6171
bP (nmol/L)	2.14 ± 1.92	2.21 ± 1.81	2.22 ± 1.93	1.89 ± 1.86	0.5809	0.1919
bPRL (ng/mL)	21.96 ± 55.79	17.64 ± 32.81	21.50 ± 53.51	23.34 ± 62.62	0.1707	0.8039
bT (nmol/L)	2.16 ± 4.85	2.14 ± 5.29	1.99 ± 4.47	2.67 ± 5.86	0.9667	0.2908
Gn times (days)	11.00 ± 2.09	10.82 ± 2.31	10.93 ± 2.06	11.22 ± 2.15	0.2644	0.2805
Gn doses (IU)	2506 ± 854.2	2551 ± 893.8	2480 ± 853.8	2586 ± 856.2	0.4821	0.3486
ICSI (%)	26.62 (82/308)	24.38 (118/484)	25.43 (59/232)	30.26 (23/76)	0.4787	0.4081
Number of oocytes retrieved	13.15 ± 6.87	11.46 ± 6.61	13.01 ± 6.84	13.55 ± 6.98	0.0006 *	0.5529
Fertilization rate (%)	82.80 (3134/3785)	79.52 (4167/5240)	82.88 (2348/2833)	82.56 (786/952)	<0.0001 *	0.8224
Cleavage rate (%)	98.15 (3076/3134)	97.98 (4083/4167)	97.96 (2300/2348)	98.73 (776/786)	0.6130	0.1645
High-quality embryo rate (%)	68.16 (2136/3134)	66.79 (2783/4167)	67.63 (1588/2348)	69.72 (548/786)	0.2166	0.2767
Blastocyst formation rate (%)	54.45 (1675/3076)	54.27 (2216/4083)	54.91 (1263/2300)	53.09 (412/776)	0.8797	0.3786
High-quality blastocyst rate (%)	41.09 (1264/3076)	39.43 (1610/4083)	41.39 (952/2300)	40.21 (312/776)	0.1559	0.5618
At least one high-quality blastocyst rate (%)	97.73 (301/308)	87.19 (422/484)	98.28 (228/232)	96.05 (73/76)	<0.0001 *	0.4932
Endometrial thickness (ET day) (mm)	10.53 ± 1.77	10.36 ± 2.78	10.53 ± 1.75	10.54 ± 1.86	0.3260	0.9832
SBT (%)	43.51 (134/308)	53.72 (260/484)	49.14 (114/232)	26.32 (20/76)	0.0051 *	0.0005 *

MS: medium age with successful birth group; MU: medium age with unsuccessful birth; MF: medium age with full-term birth; MP: medium age with preterm birth; BMI, body mass index; bFSH: basic follicle-stimulating hormone; bLH: basic luteinizing hormone; bE2: basic estradiol; bP: basic progesterone; bPRL: basic prolactin; bT: basic testosterone; Gn: gonadotropins; ICSI: intracytoplasmic single sperm injection; SBT: single blastocyst transfer. Measurement data are presented as means ± standard deviations (SD); comparisons between two groups were performed using the *t*-test. Enumeration data were reported as percentages and compared by using the χ2 test, *: *p* < 0.05, difference was statistically significant.

**Table 4 jcm-12-01618-t004:** Multivariate logistic regression analysis of possible correlates of live birth and preterm birth in the middle group (women aged between 35 and 39).

	Index	Odds Ratio	95% Conf. Interval	*p*-Value
**Live birth**				
	Primary infertility	1.34	0.953–1.884	0.093
	Number of oocytes retrieved	1.03	1.008–1.053	0.008 *
	83% fertilization rate	1.332	0.953–1.884	0.093
	At least one high-quality blastocyst	5.424	2.422–12.147	<0.001 *
	SBT	1.482	1.099–1.999	0.01 *
**Preterm birth**				
	SBT	2.705	1527–4.792	0.001 *

SBT: single blastocyst transfer; 83% fertilization rate: univariate analysis in the medium group (women aged between 35 and 39) with live birth found that significant differences between the MS and MU groups in the rate of fertilization; therefore, the mean fertilization rate of 83% in the MS group was selected as a covariate in the multivariate logistic regression, *: *p* < 0.05, difference was statistically significant.

**Table 5 jcm-12-01618-t005:** Cycle outcomes of the four study groups.

Index	(G) LSBT	(H) LDBT	(I) HSBT	(J) HDBT	*p*-Value (G vs. H)	*p*-Value (I vs. J)
Total cycles (n)	167	156	194	205	-	-
Biochemical pregnancy rate (%)	46.71 (78/167)	62.18 (97/156)	59.28 (115/194)	69.27 (142/205)	0.0053 *	0.0372 *
Clinical pregnancy rate (%)	42.51 (71/167)	53.85 (84/156)	55.15 (107/194)	64.88 (133/205)	0.0417 *	0.0474 *
Multiple gestational sacs rate (%)	0 (0/69)	32.53 (27/83)	1.89 (2/106)	41.54 (54/130)	<0.0001 *	<0.0001 *
Live birth rate (%)	28.74 (48/167)	42.95 (67/156)	42.78 (83/194)	50.73 (104/205)	0.0077 *	0.1118
Multiple birth rate (%)	2.08 (1/48)	11.94 (8/67)	3.61 (3/83)	31.73 (33/104)	0.1121	<0.0001 *
Preterm birth rate (%)	20.83 (10/48)	23.88 (16/67)	12.05 (10/83)	36.54 (38/104)	0.7001	0.0001 *

L: number of oocytes retrieved ≤ 10; H: number of oocytes retrieved > 10; SBT: single blastocyst transfer; DBT: double blastocyst transfer, *: *p* < 0.05, difference was statistically significant.

## Data Availability

The data underlying this article will be shared upon reasonable request with the corresponding author.

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
