# Peer review of "Blastocyst Transplantation Strategies in Women of Different Ages"

_jcm, 2023, doi:10.3390/jcm12041618_

Round 1

Reviewer 1 Report

This comprehensive and retrospective study provided valuable information on SBT and DBT use in clinical practice. 

Please correct line 35: "in the past days" with the appropriate timing (perhaps years?). 

Question for the authors: Did you find any correlations between your data and PGT/PGD results? 

Author Response

We thank the reviewer for careful review and insightful comments on our manuscript entitled “Blastocyst transplantation strategies in women of different ages” (Manuscript ID: jcm-2163034).

The original manuscript has been carefully revised based on the reviewer’s comments and concerns. The following is the point-by-point response.

  1. Thank you for your comment, we have corrected the manuscript in the appropriate places: correct “However, multiple embryos were transferred to increase the clinical pregnancy rate and live birth rate per embryo transfer cycle in the past days” (line 34-35) to “However, multiple embryos were transferred to increase the clinical pregnancy rate and live birth rate per embryo transfer cycle over the past several decades.” (line 34-35).
  2. Thank you for your suggestion. In fact, to avoid sample selection bias, PGD/PGT cycles were not included in this study. Because we clinically transplanted only single blastocyst that was biopsied for patients in PGD/PGT cycles. In future studies, we will try to explore the transplantation strategy in PGD/PGT patients with reference to your suggestion.

Reviewer 2 Report

1.      Line 68. A total of 5477 cycles were enrolled

How many patients are included in this study? There are repeat cycles?

2.      Line 77. And oocyte retrieval was completed after 24–36h by a transvaginal ultrasound.

The correct time of oocyte retrieval should be explained.

3.      Statistical analysis: In this study, t-test was used to analyze two independent samples. Are these data normally distributed?

4.      Statistical analysis: The day of transferred blastocysts is not analyzed in this study.  The day and quality of transferred blastocyst should be included as a confounding factor for pregnancy outcomes in regression analyses.

5. Statistical analysis: If repeated cycles were present in this study, GEE regression should be an appropriate method of analysis.
